# Symmetry Analysis of Magnetoelectric Effects in Perovskite-Based Multiferroics

**DOI:** 10.3390/ma15020574

**Published:** 2022-01-13

**Authors:** Zukhra Gareeva, Anatoly Zvezdin, Konstantin Zvezdin, Xiangming Chen

**Affiliations:** 1Institute of Molecule and Crystal Physics, Subdivision of the Ufa Federal Research Centre of the Russian Academy of Sciences, Prospect Octyabrya 151, 450075 Ufa, Russia; 2Institute of Physics and Technology, Bashkir State University, ul. Z. Validi 32, 450076 Ufa, Russia; 3Prokhorov General Physics Institute of the Russian Academy of Sciences, ul. Vavilova 38, 119991 Moscow, Russia; zvezdin.ak@phystech.edu (A.Z.); zvezdin.ka@phystech.edu (K.Z.); 4Laboratory of Dielectric Materials, School of Materials Science and Engineering, Zhejiang University, Zheda Road 38, Hangzhou 310027, China; xmchen59@zju.edu.cn

**Keywords:** keyword multiferroics, crystal lattices, symmetry, perovskites, magnetoelectric effect

## Abstract

In this article, we performed symmetry analysis of perovskite-based multiferroics: bismuth ferrite (BiFeO_3_)-like, orthochromites (RCrO_3_), and Ruddlesden–Popper perovskites (Ca_3_Mn_2_O_7_-like), being the typical representatives of multiferroics of the trigonal, orthorhombic, and tetragonal crystal families, and we explored the effect of crystallographic distortions on magnetoelectric properties. We determined the principal order parameters for each of the considered structures and obtained their invariant combinations consistent with the particular symmetry. This approach allowed us to analyze the features of the magnetoelectric effect observed during structural phase transitions in Bi_x_R_1−x_FeO_3_ compounds and to show that the rare-earth sublattice has an impact on the linear magnetoelectric effect allowed by the symmetry of the new structure. It was shown that the magnetoelectric properties of orthochromites are attributed to the couplings between the magnetic and electric dipole moments arising near Cr^3+^ ions due to distortions linked with rotations and deformations of the CrO_6_ octahedra. For the first time, such a symmetry consideration was implemented in the analysis of the Ruddlesden–Popper structures, which demonstrates the possibility of realizing the magnetoelectric effect in the Ruddlesden–Popper phases containing magnetically active cations, and allows the estimation of the conditions required for its optimization.

## 1. Introduction

Multiferroic, multifunctional materials attract enormous research activity due to the prospects of their implementation in science and technology. The collective state parameters attainable in multiferroics (MFs) are preferable for advanced spintronic devices based on the use of combinatorial functions due to their high efficiency, low power consumption, and adaptability [1].

Despite the variety of currently known magnetoelectric materials, perovskite-like structures remain the most researched MFs compounds. The family of perovskites stands out for its diversity due to the intrinsic instability of the cubic parent perovskite phase [2]. Crystallographic distortions owing to the substitution of various cations at A and B positions can lead to the emergence of ferroelectric and MF properties due to the hybridization of ion orbitals, tilt of oxygen octahedra (BO_6_), doping of various cations, and other factors [3]. ABO_3_ perovskites can produce a variety of MFs differing in the type of (i) ferroelectric ordering (A- or B-driven ferroelectricity), as in the cases of EuTiO_3_ [4] and BiFeO_3_ [5]; (ii) magnetic ordering (although most perovskite MFs are G-type antiferromagnets (AFM), canted ferromagnetism and a combination of ferromagnetic and AFM phases are also possible) [6,7]; (iii) mechanism of magnetoelectric coupling, which can be associated with magnetoelectrically active d-ions or f-ions (RFeO_3_) [8], the Dzyaloshinskii–Moriya interaction (BiFeO_3_) [5], spin–orbit coupling (BiMnTe) [9], antiferroelectric orderings (RCrO_3_)) [10]; and (iv) temperatures of ferroelectric and magnetic orderings, which remain low enough for most perovskite-based MFs.

Quite recently, the MF community turned to hybrid layered perovskite structures, considering them as prospective candidates for high-temperature MFs. We refer to layered structures composed of an n-layered perovskite with a single-layer spacer in-between. Depending on the kind of single-layer spacer, one can distinguish the Ruddlesden–Popper (RP) A′_2_(A_n−1_B_n_O_3n+1_), Dion–Jacobson (DJ) A′(A_n−1_B_n_O_3n+1_), and Aurivillius Bi_2_O_2_(A_n−1_B_n_O_3n+1_) structures [11,12]. When n = ∞, the layered perovskite transforms into a classical perovskite.

Among layered perovskites, the most widespread and studied compounds are the RP structures. They received their name in honor of the chemists R. Ruddlesden and P. Popper, who, in 1957, synthesized the series of Sr_2_TiO_4_, Sr_3_Ti_2_O_7_, and Sr_4_Ti_3_O_10_ phases, forming a class of complex oxides S_rn+1_Ti_n_O_3n+1_ [13]. Since then, the class of RP structures has expanded significantly, and, at present, it includes various compounds with the general formula A′_2_(A_n−1_B_n_O_3n+1_), where A’ = La, Sr, Li, …; A = Sr, Ca, K, …; and B = Ti, Fe, Mn, Nb, …. The advantages of RP-type perovskites are related to long-term chemical stability, the activity of oxygen reduction reaction, and the possibility of realizing nontrivial magnetic and electrical properties [11,12,13,14,15,16,17]. Due to the activity of the reaction with oxygen, RP structures find applications in solid oxide fuel cells [14]. As for magnetoelectricity, interest in layered perovskites is associated with the discoveries of the colossal magnetoresistance effect in the early 2000s in RP structures with magnetoactive cations (Ca_3_Mn_2_O_7_, …) [14,15,16,17,18,19] and ferroelectricity at room temperatures, found in Ca_3−x_Sr_x_Ti_2_O_7_ in 2015 [20] and then in Sr_3_M_2_O_7_-based oxides and other RP phases [21]. Currently, RP structures are considered promising candidates for high-temperature MFs. The crystal structure, instabilities, and structural phase transitions occurring in RP oxides are well known; however, a number of questions concerning ferroelectricity, magnetic ordering, and their coupling is open for discussion.

The focus of this manuscript was the study of the magnetoelectric effect in Bi_1−x_R_x_FeO_3_, RCrO_3_, and Ca_3_(Ti_1−x_Mn_x_)_2_O_7_ multiferroics (MFs). We aimed to develop a unified approach to the analysis of the perovskite-like multiferroics based on the symmetry of the material, taking crystallographic distortions as the primary order parameters.

It is valuable to note that the classification of distortions and the search for recipes for effective magnetoelectric couplings in MFs with a perovskite structure is a long-standing and, at the same time, ‘hot’ problem. We refer to several reviews, references therein, and original papers that discussed various classification schemes for perovskite distortions and group-theoretical approaches, including their implementation in online tools (in particular, ones on the Bilbao Crystallographic Server) for identification of active order parameters, and their possible couplings and invariant polynomials [22,23,24,25,26,27,28,29]. However, distortion classification schemes mainly deal with BO6 octahedrons and do not account for polar distortions, which have significant impact on ferroelectric and MF properties; in addition, they also have limitations for specific systems. Despite the power of the software tools, they have not yet been applied to each MF system; for example, the details of the magnetoelectric coupling in MFs with a variable concentration of rare-earth ions have not yet been investigated, and the same statement is also applied to the RP structures.

Thus, in our research, we appealed to group-theoretical analysis, which is an elegant and effective tool for studying the properties of crystals with a complex magnetic structure, to classify the distortive, ferroelectric, and magnetic orderings in several classes of perovskite MFs. For this purpose, we examined typical MF structures with the perovskite parent phase: BiFeO_3_, RCrO_3_, Ca_3_Mn_2_O_7_, in which small distortions in the crystallographic lattice lead to systems of different symmetry (trigonal, orthorhombic, and tetragonal). Using the methods of theoretical group analysis, we considered how the symmetry manifests itself in their magnetoelectric properties. We determined magnetic and structural modes, classified them according to the irreducible representations of the corresponding symmetry group, and calculated the coefficients of magnetoelectric couplings.

## 2. Materials and Methods

In this section, we analyze the crystal structure, and ferroelectric and magnetic properties of representatives of the perovskite-based multiferroics and, using the methods of the theoretical group analysis, determine the features of their magnetoelectric properties.

### 2.1. Multiferroic BiFeO_3_

We start our consideration with the well-known multiferroic BiFeO_3_ with high temperatures of ferroelectric and magnetic orderings (T_C_ = 1083 K, T_N_ = 643 K). BiFeO_3_ crystallizes in the nonpolar symmetry group *R*3*c* and is a ferroelectric material. The high-symmetry perovskite group is reduced to *R*3*c* due to three types of distortions: (i) relative displacement of Bi and Fe ions along <111> axis, (ii) deformations of oxygen octahedrons, and (iii) counterrotation of oxygen octahedrons around Fe ions. The displacements of Bi and Fe ions from their centrosymmetric positions lead to spontaneous electric polarization ***P_s_*** directed along one of the <111> crystallographic axes [30]. The magnitude of polarization has been the subject of controversy for a while. As was reported in Refs. [5,20,31,32,33,34,35], spontaneous polarization in the BiFeO_3_ crystal and films can attain values varying from 6 to 150 μC/cm^2^. Electric polarization is sufficiently low, ***P_s_***~6–9 μC/cm^2^, in single crystals and ceramic samples [34,35]; the large values of ***P_s_***, around 100 μC/cm^2^, are achieved under epitaxial strain in the films [5,36,37,38]; and they can also be related to the supertetragonal phase induced by the strain.

The rigorous estimation of electronic polarization requires the implementation of quantum mechanics, including electronic structure methods with correspondence to experimentally measurable observables [39]. In the frame of these approaches in Ref. [40], electric polarization in perovskite BiFeO_3_ was calculated as a function of percentage distortion from the high-symmetry nonpolar structure to the ground-state *R*3*c* structure, which gives the value 95.0 μC cm^−2^. However, as was shown in Ref. [30], similar results can be obtained with the use of the point charge model, where the electric polarization is represented through the atomic displacements of Bi^3+^ and O^2−^ ions.
(1)$$P=\frac{4e}{V}\left(3\zeta_{R}-4\zeta_{O1}-2\zeta_{O3}\right)$$
where *e* is the elementary charge, *V* is the unit cell volume, *ζ_R_* is the displacement of Bi ions corresponding to (i) distortion, and ζ_O1_ and ζ_O3_ are the displacements of oxygen ions corresponding to (ii)—distortion.

In magnetic relation, BiFeO_3_ is weak ferromagnet with G-type antiferromagnetic (AFM) ordering characterized by AFM vector $L=\frac{1}{V}{\sum_{i=1}^{6}\left(-1\right)}^{i}\mu_{i}$, where ***μ****_i_* are the magnetic moments of six Fe ions in a unit cell. The canting of spins is related to rotations of oxygen octahedrons FeO6, determined by the vector **Ω**, the distortions of (iii)—type [41], which have an impact on the weak ferromagnetic vector, determined as
(2)$$M=\frac{V_{0}a^{2}}{6J}\Omega \times L+\frac{V_{0}a^{2}}{6J}\sum_{n=1}^{6}\left(n_{n}\times \zeta_{O_{n}}\right)$$
where *V*_0_ is a constant, *a* is the lattice parameter, *J* is the exchange constant, and ***n**_n_* is the direction vector, oriented along one of the <100> crystallographic axes. Here, we used an assumption that the Heisenberg exchange constant remains unchanged, which is a commonly used approximation for multiferroic BiFeO_3_; in particular, the deviation (noncollinearity) of the nearest-neighbor spins is accounted for by the antisymmetric Dzyaloshinskii–Moriya exchange interaction [42,43]. AFM spin arrangement is superimposed with a spatially modulated magnetic structure with a large period λ = (620 ± 20 Å) incommensurate with the lattice parameter [44]. The magnetic moments of Fe ions retain their local mutually antiferromagnetic G-type orientation and rotate along the propagation direction of the modulation in the plane perpendicular to the basal crystal plane.

To explore the magnetoelectric effect (MEE) in BiFeO_3_, we appealed to symmetry analysis. Knowledge of crystal structure and symmetry is of primary importance for this approach. To consider the properties of BiFeO_3_, the space group $R3\bar{c}$ was taken as a ‘parent’ symmetry phase [45]. $R3\bar{c}$ differs from the space symmetry group *R*3*c* of BiFeO_3_ single crystals only by the presence of a polar vector ***P*** = (0, 0, *P_s_*). The symmetry analysis of the magnetoelectric properties of BiFeO_3_ has been performed in Refs. [45,46,47]. Here, we focused our attention on the magnetoelectric tensor *α_ij_*, which links the electric polarization and the magnetic field
(3)$$P_{i}=\alpha_{ij}H_{j}$$

As was shown in Ref. [47] for BiFeO_3_ crystals, the tensor *α_ij_* can be expressed through the components of the AFM vector ***L***
(4)$$\alpha_{ij}^{BiFeO_{3}}=\left|\begin{array}{l} -a_{1}L_{x}\;a_{4}L_{z}+a_{1}L_{y}\;a_{2}L_{y}\\ a_{1}L_{y}-a_{4}L_{z}\;a_{1}L_{x}\;-a_{2}L_{x}\\ -a_{3}L_{y}\;a_{3}L_{x}\;0 \end{array}\right|$$

As seen from Equations (3) and (4), the symmetry allows the linear magnetoelectric effect; however, due to the presence of an incommensurate space-modulated structure, it is not observed in BiFeO_3_. Thus, when the spin cycloidal structure is destroyed, the linear magnetoelectric effect is restored. It can be achieved by applying a strong magnetic field, pressure, or doping BiFeO_3_ with rare-earth ions. In recent years, a series of experimental studies has been conducted, indicating the enhancement of the magnetoelectric effect in BiFeO_3_ multilayers and composites.

Here, we discuss the replacement of various rare-earth (RE) ions at the A-positions as a way to improve the MF properties of BiFeO_3_, a route that was paved in the 1990s [30,45,46,47,48,49,50], and current research confirms its efficiency [6,51]. As was shown in Ref. [51], La-substitution in Bi_1−x_La_x_FeO_3_ (x = 0.08~0.22) ceramics induces a continuous structural evolution from *R*3*c* to *Pna*2_1_ and, finally, the *Pbnm* phase. To understand the properties of Bi_1−x_R_x_FeO_3_ structures (R = La, Gd, Dy, …), it is reasonable to introduce the AFM vector (***l***), which characterizes the magnetic ordering of the rare-earth sublattice, and to consider the *Pbnm* space symmetry group as the parent phase in the Bi_1−x_R_x_FeO_3_ structure, keeping in mind structural phase transitions *Pbnm* → *Pna*2_1_. The space group *Pbnm* contains eight irreducible representations, where their matrix representations are given in the columns corresponding to the generating symmetry elements (Table 1). The vector components of magnetic field ***H***, electric polarization ***P***, magnetization ***M***, and antiferromagnetic vectors ***L*** and ***l*** form the irreducible representations and are placed in the last column in Table 1 according to their transformation properties. Table 1 allows the determination of the transformation properties of AFM vector components, exchange-coupled structures, and magnetoelectric invariants, contributing to the magnetoelectric energy of the Bi_1−x_R_x_FeO_3_ structure.
(5)$$\begin{array}{l} \Phi_{me}^{B_{1-x}R_{x}FeO_{3}}=P_{z}\left(\gamma_{1}\left(l_{z}M_{y}-l_{y}M_{z}\right)+\gamma_{2}\left(l_{x}L_{y}-l_{y}L_{x}\right)\right)+P_{x}\left(\gamma_{xyx}\left(l_{y}M_{x}-l_{x}M_{y}\right)+\gamma_{2}\left(l_{y}L_{z}-l_{z}L_{y}\right)\right)+\\ P_{y}\left(\gamma_{3}l_{x}M_{x}+\gamma_{2}l_{x}L_{z}+\gamma_{4}\left(l_{y}M_{y}+L_{y}^{2}\right)\right) \end{array}$$
where *γ_i_* are the magnetoelectric coefficients,
$$\gamma_{1}=\gamma_{zzy},\gamma_{2}=\gamma_{xyz},\gamma_{3}=\gamma_{yxx},\gamma_{4}=\gamma_{yyy}$$

The magnetoelectric coupling coefficient for the Bi_1−x_R_x_FeO_3_ structures can be determined as follows
(6)$$\alpha_{ij}=\left|\begin{array}{l} \;a_{1}\left(l_{x}L_{x}+l_{z}L_{z}\right)\;a_{12}l_{x}\;-a_{13}l_{y}\;\\ \;a_{12}l_{z}\;a_{2}l_{y}\;a_{23}l_{x}\;\\ \;a_{13}l_{y}\;a_{32}l_{z}\;a_{1}\left(l_{x}L_{x}+l_{z}L_{z}\right) \end{array}\right|\approx \left|\begin{array}{l} \;0\;a_{12}l_{x}\;-a_{13}l_{y}\;\\ \;a_{12}l_{z}\;a_{2}l_{y}\;a_{23}l_{x}\\ \;a_{13}l_{y}\;a_{32}l_{z}\;0 \end{array}\right|$$

In the case of Bi_1−x_R_x_FeO_3_, rare-earth elements can have a significant impact on magnetoelectric properties. As seen from Equation (6), *α_ij_* depends on the components of the AFM vector of the rare-earth sublattice (for the case of nonzero magnetization on the R ion site), and it is possible to distinguish a linear MEE related to rare-earth elements, and a quadratic MEE emerges due to cross-coupling of the AFM vectors of iron and rare-earth sublattices.

### 2.2. Magnetoelectric Properties of RCrO_3_

In this section, we consider MEE in the rare-earth orthochromites (RCrO_3_), focusing on the relation of magnetoelectric properties with crystallographic distortions. The crystal structure of RCrO_3_ belongs to the space symmetry group *Pbnm/Pnma* [52,53,54]. Its orthorombically distorted perovskite unit cell contains 4 RCrO_3_ molecules (Figure 1). In addition, 4 Cr^3+^ and 4 R^3+^ magnetic ions (in the case of R = rare-earth ion) are located in the local positions differing by the symmetry of O^2−^ environments. The *d-*ions (M^3+^) occupy the position 4*b*, the *f*-ions (R^3+^) occupy the position 4*c*, and oxygen ions occupy the positions 4*c* and 8*d* (in Wyckoff notation).

Magnetic moments of the *d-*ions determined by the vectors ***M****_i_* (*i* = 1–4) constitute 4 transition metal magnetic sublattices, and the magnetic moments of the *f*-ions determined by vectors *m_i_* (*i* = 1–4) constitute 4 rare-earth magnetic sublattices. The combinations between magnetic moments of the *d*- and the *f*-sublattices determine magnetic modes of the ‘*d*’ ions:$$F=M_{1}+M_{2}+M_{3}+M_{4},\;A=M_{1}-M_{2}-M_{3}+M_{4}$$
(7)$$G=M_{1}-M_{2}+M_{3}-M_{4},C=M_{1}+M_{2}-M_{3}-M_{4}$$
and magnetic modes of the ‘*f*’- ions:(8)$$\begin{array}{l} f=m_{1}+m_{2}+m_{3}+m_{4},\;a=m_{1}-m_{2}-m_{3}+m_{4},\;\\ g=m_{1}-m_{2}+m_{3}-m_{4},\;c=m_{1}+m_{2}-m_{3}-m_{4} \end{array}$$

Neutron diffraction measurements showed that RCrO_3_ exhibits one of three *G*-type AFM configurations with weak magnetic components $\Gamma_{1}\left(A_{x},G_{y},C_{z}\right)$, Г_2_ (*F_x_,C_y_,G_z_*), and Г_4_ (*G_x_A_y_F_z_*).

In contrast to BiFeO_3_, the ferroelectric properties of RCrO_3_ are weak, and the Curie ordering temperatures are low (T_C_~130–250 K) [55,56]. However, as recent experiments have shown, during the polishing of samples in an electric field, an electric polarization of the order of 0.06 μC/cm^2^ is induced; moreover, the electric field affects the spin–reorientation phase transitions, which indicates the presence of magnetoelectric coupling in the material [57,58].

As expected, the magnetoelectric behavior is owed to the crystallographic distortions, which are quite different from those in BiFeO_3._ In the case of RCrO_3_, the principal distortions are related to the antirotation of the CrO6 octahedrons around the [110] direction and the displacements of R ions from A-positions in the perovskite parent phase. Octahedron rotation is described by the axial vectors ***ω****_i_* (*i* = 1 ÷ 4) linked to Cr ions, and, as was shown in Ref. [10], they contribute to the axial order parameter.
(9)$$\Omega_{b}=\omega_{1}-\omega_{2}+\omega_{3}-\omega_{4}$$

Polar order parameters related to the displacements of oxygen and rare-earth ions are also allowed. To compose the polar order parameters, we consider the electric dipole moments $d_{i}=qr_{i}$ emerging in the vicinity of the Cr ions, where $r_{q}=\sum_{i}^{}q_{i}r_{qi}^{}/\sum_{i}^{}q_{i}$ is the position of the electric dipole charge center, *q_i_* are the signed magnitudes of the charges, and ***r****_qi_* are the radius vectors of the charges in the local reference frame. For the perovskite-like compounds,
(10)$$r_{q}=\frac{\left(+\frac{3}{8}e\right)\cdot \sum_{i=1}^{8}r_{R}+\left(-\frac{2}{2}e\right)\cdot \sum_{i=1}^{6}r_{O}}{\left|8\cdot \left(+\frac{3}{8}e\right)+6\cdot \left(-\frac{2}{2}e\right)\right|}\;r=(x,y,z)$$
where *e* is the elementary charge, ***r****_R_* are the radius vectors of the rare-earth ions, and ***r****_O_* are the radius vectors of the oxygen ions measured from the Cr^3+^ ion.

As in the case of magnetic moments (7), (8), electric dipoles $d_{i}=qr_{i}$ constitute 4 ferroelectric sublattices, and the combinations between dipole moments determine ferroelectric modes reads.
(11)$$\begin{array}{l} P=d_{1}+d_{2}+d_{3}+d_{4}\\ Q_{2}=d_{1}-d_{2}-d_{3}+d_{4}\\ Q_{3}=d_{1}-d_{2}+d_{3}-d_{4}\\ D=d_{1}+d_{2}-d_{3}-d_{4} \end{array}$$

The arrangement of the dipole moments in RCrO_3_ is shown in Figure 1; as can be seen, they form an antiferroelectric structure ordered according to the ***D***-mode. Direct calculation of the dipole moments shows that the value of ***D*** is maximum, and the values of ***P*, *Q***_2,3_ are negligible.

The transformation properties of the order parameters introduced above can be found using the irreducible representations of the space symmetry group of the RCrO_3_ compounds given in Table 2.

Knowledge of their transformation properties makes it possible to understand how crystallographic distortions manifest themselves in magnetic and ferroelectric orderings. As in the previous subsection, classification of the components of order parameters according to the irreducible representations of the *Pnma* symmetry group allows us to determine exchange-coupled magnetic structures, the possible electrostatically coupled dipole structures, to compose the invariant combination of order parameters contributing to the thermodynamic potential and to calculate the magnetoelectric coupling coefficient.
*P_x_ = (α_xyx_a_y_ + α_xzyx_g_z_G_y_G_x_ + α_xxyx_a_x_G_y_ G_x_)H_x_ + (α_xxy_a_x_ + α_xxyy_f_x_G_y_ G_x_ + α_xzyy_c_z_G_y_ G_x_ + α_xzy_g_z_)H_y_ +…*
*F_x_ = (α*_xyx_a_y_ + α*_xzyx_g_z_G_y_ G_x_ + α*_xxyx_a_x_G_y_ G_x_)E_x_ + (α*_xxy_a_x_ + α*_xxyy_f_x_G_y_ G_x_ + α*_xzyy_c_z_G_y_G_x_ + α*_xzy_g_z_)E_y_ +…*(12)

The magnetoelectric coefficient in terms of the components of ***G*** and ***g*** vectors is written as follows.
(13)$$\alpha_{ij}^{RCrO_{3}}=\left|\begin{array}{l} a_{1}G_{x}\left(g_{z}G_{y}-g_{y}G_{z}\right)\;a_{2}g_{z}\;a_{3}g_{x}\\ a_{2}g_{z}\;a_{1}G_{x}\left(g_{z}G_{y}-g_{y}G_{z}\right)\;a_{4}g_{y}\\ a_{4}g_{y}\;a_{3}g_{x}\;a_{2}g_{z} \end{array}\right|$$

At the end of this section, we draw the reader’s attention to the fact that the presented analysis applies to rare-earth orthoferrites RFeO_3_. RFeO_3_ compounds crystallize in the same space symmetry group *Pnma*, and several of these crystals exhibit nonlinear magnetoelectric responses [23,25,59]. The magnetoelectric tensor (12) can also be used to describe their magnetoelectric properties.

### 2.3. Ruddlesden–Popper Structures

Ruddlesden–Popper (RP) structures are attracting considerable attention due to the recently discovered high-temperature ferroelectric properties and the prospects for their potential implementation as multiferroics at room temperature [11]. The presence of ferroelectric properties, the so-called hybrid improper ferroelectricity (HIF), was theoretically predicted in double-layered perovskite compounds Ca_3_Mn_2_O_7_ and Ca_3_Ti_2_O_7_, and then detected experimentally in Ca_3−x_Sr_x_Ti_2_O_7_ single crystals [20], and in Ca_3_(Ti_1−x_Mn_x_)_2_O_7_ ceramics [21].

In this section, we plan to focus on the RP manganite phases Ca_3_(Ti_1−x_Mn_x_)_2_O_7_ as they contain magnetic Mn ions and, as shown in Ref. [21], are room-temperature HIF materials. In the limiting case x = 1, Ca_3_(Ti_1−x_Mn_x_)_2_O_7_ transforms into the RP phase Ca_3_Mn_2_O_7_, which was thoroughly explored due to the colossal magnetoresistance effect found in the (CaO)-(CaMnO_3_)_n_ (n =1, 2, 3, ∞) RP structures in the 2000s [16,17,18,19]. The structures with n = 1 (2D-structure) and n = ∞ (perovskite) are considered as the end members of the RP series, so the double-layered RP structures receive more attention.

The temperature of AFM ordering T_N_ = 115 K in Ca_3_Mn_2_O_7_ was first experimentally found in Ref. [16], where it was assumed that there is a G-type AFM order. A neutron diffraction study [19] revealed the dominant contribution of the G-type or C-type AFM phase with an AFM spin arrangement within the bi-layer plane. In that work, the possibility of the existence of a weak ferromagnetic (WFM) state in the RP phase Ca_3_Mn_2_O_7_ was also assumed. A further study on Ca_3_Mn_2_O_7_ showed the transition to the AFM state at the temperature T_N_ = 134 K and a signature of WFM below 100 K [60,61].

The crystal structure, structural phase transitions, and polar phases of Ca_3_Ti_2_O_7_ [62] and Ca_3_Mn_2_O_7_ [19,63] with double-layered RP structures, studied since 1998 [16], have now been proven [21]. Above room temperature (RT), the crystal structure of Ca_3_Mn_2_O_7_ is described by the tetragonal space group *I*4/*mmm*, and at RT, by the space group *Cmc*2_1_ [64]. The transition from the *I*4/*mmm* to *Cmc*2_1_ phase should occur through an intermediate phase, which can be either *Cmcm* or *Cmca,* as follows from the first-principles calculations [46]. The transition into the ferroelectric phase is observed close to *T_C_* = 280 K, at which the transition into the *Cmc*2_1_ phase occurs. As experimental measurements indicate that ferroelectric ordering in Ca_3_Mn_2_O_7_ occurs up to RT, the existence of an intermediate ferroelectric state is expected here.

Structural transformations can arise due to the substitution of magnetic ions at the A-positions (Ca_3−x_La_x_Mn_2_O_7_) or ferroelectric ions in the B-positions (Ca_3_(Ti_1−x_Mn_x_)_2_O_7_). The tetragonal-to-orthorhombic transition presumable through the phases *I*4*/mmm* → *Fmmm* → *Cmcm* → *Cmc*2_1_, occurring in the 200–300 °C range, was observed during the study of Ca_3−x_La_x_Mn_2_O_7_ structures [19,64]. Though the existence of *I*4/*mmm* and *Cmc*2_1_ phases is confirmed, the presence of ‘intermediate’ phases is still under discussion.

Thus, in further consideration, we can take the tetragonal *I*4*/mmm* structure (Ca_3_Mn_2_O_7_) as the parent phase and apply the symmetry analysis, employed in the previous sections, to explore the magnetoelectric properties of the RP structures. According to the data of the synchrotron X-ray diffraction study [19], the Mn ions occupy the 4e positions (in the Wyckoff notation) in the *I*4*/mmm* phase, which indicates the presence of 4 magnetic sublattices ***μ_i_*** (*i* = 1÷4) with magnetic moments of the same magnitude. Possible combinations of the vectors ***μ_i_*** give us the basic magnetic vectors.
(14)$$\begin{array}{l} F=\mu_{1}+\mu_{2}+\mu_{3}+\mu_{4}\\ A=\mu_{1}-\mu_{2}-\mu_{3}+\mu_{4}\\ G=\mu_{1}-\mu_{2}+\mu_{3}-\mu_{4}\\ C=\mu_{1}+\mu_{2}-\mu_{3}-\mu_{4} \end{array}$$

The unit cell of Ca_3_Mn_2_O_7_ shown in Figure 2 contains 2 formula units Ca_3_Mn_2_O_7_. The positions of the symmetry elements $I,C_{2}^{z},C_{4}^{z},C_{2}^{y}\;$(generators of *I*4/*mmm* space group) in the unit cell are depicted in Figure 2b in accordance with Figure 2a. Note that the local symmetry elements of the 4e{4mm} positions, which are the $4_{z}$ axis (principal crystal axis) and 4 planes of symmetry *m* passing through this axis, remain even for magnetic ions, occupying the 4e positions.

In accordance with Figure 2, it is easy to compose permutation transformations of the ions at the 4e positions (Table 3) and permutation transformations of basis vectors (Table 4).

Using Table 4, we obtain a cipher (Turov indices) for AFM vectors, which explains how the symmetry elements transform magnetic sublattices into each other.
$$A:\;4_{z}\left(+\right)2_{z}^{1}\left(-\right)2_{y}\left(-\right)\bar{1}\left(-\right)\;$$
$$G:\;4_{z}\left(+\right)2_{z}^{1}\left(+\right)2_{y}\left(-\right)\bar{1}\left(-\right)\;$$
$$C:\;4_{z}\left(+\right)2_{z}^{1}\left(-\right)2_{y}\left(+\right)\bar{1}\left(+\right)\;$$

In consistence with the data of experimental studies [16,19], we assume the G-type of AFM ordering. In this case, the values of AFM vectors ***A*** and ***C*** are taken to be sufficiently small and we can restrict our consideration with two magnetic order parameters instead of 4. Thus, we leave the ferromagnetic vector ***M*** = ***F*** and the antiferromagnetic vector ***L*** = ***G*** as the basic magnetic parameters. Then, we classify them together with the polarization vector according to the irreducible representation of the space symmetry group $D_{4h}^{17}$ (Table 5). The first line of Table 5 contains generators of the $D_{4h}^{17}$ group $G=\left\{E,I,C_{2}^{y},C_{2}^{z},C_{4}^{z+}\right\}$, where the last column contains basic magnetic and ferroelectric functions.

The decomposition of the basic functions into irreducible representations of the symmetry group *I*4/*mmm* allows one to obtain information on the properties of a system. It is seen that the ferromagnetic and antiferromagnetic vectors transform according to the different IRs. In this case the transformation of ***F*** and ***L*** into each other can occur due to external factors, such as the electric field or strain. The combinations $F_{z}\left(P_{x}G_{x}+P_{y}G_{y}\right),G_{z}\left(F_{x}P_{x}+F_{y}P_{y}\right),P_{z}\left(F_{x}G_{x}+F_{y}P_{z}G_{y}\right),F_{z}P_{z}G_{z}$, transforming on Г_1_, are invariants and have an impact on magnetoelectric energy.
(15)$$\Phi_{me}=\gamma_{2}F_{z}\left(P_{x}G_{x}+P_{y}G_{y}\right)+\gamma_{3}G_{z}\left(F_{x}P_{x}+F_{y}P_{y}\right)+\gamma_{4}P_{z}\left(F_{x}G_{x}+F_{y}G_{y}\right)+\gamma_{5}F_{z}P_{z}G_{z}$$
where
$$\gamma_{2}=\gamma_{zxx}=\gamma_{zyy},\gamma_{3}=\gamma_{xxz}=\gamma_{yyz},\gamma_{4}=\gamma_{xzx}=\gamma_{yzy},\gamma_{5}=\gamma_{zzz}$$

Using Table 5 we calculate the magnetoelectric coupling tensor.
(16)$$\alpha_{ij}^{R-P}=\left|\begin{array}{l} \gamma_{3}G_{z}\;0\;\gamma_{2}G_{x}\\ 0\;\gamma_{3}G_{z}\;\gamma_{2}G_{y}\\ \gamma_{4}G_{x}\;\gamma_{4}G_{y}\;\gamma_{5}G_{z} \end{array}\right|$$

As seen from Equation (16) it linearly depends on the components of antiferromagnetic vector ***G***. This finding allows us to conclude that the linear magnetoelectric effect is allowed by the symmetry of the RP structures.

## 3. Results and Conclusions

To summarize, we applied symmetry analysis to the magnetic and magnetoelectric properties of multiferroics (MFs) with a perovskite structure. One of the reasons for choosing perovskite-based MFs as the object of this study is related to the structural instability of the initial perovskite phase, which can lead to a variety of structures with different symmetry. This allows us to demonstrate how crystallographic distortions of various types, even if they are small enough, significantly alter the magnetic and ferroelectric orderings and their couplings. As the typical examples, we considered (i) the family of high-temperature multiferroics Bi_x_R_1−x_FeO_3_ recognizable by their magnetoelectric properties; (ii) the rare-earth orthochromites RCrO_3_, promising candidates for MFs, compounds with well-known magnetic structures; and (iii) the RP structures containing magnetic cations, such as Ca_3_(Ti_1−x_Mn_x_)_2_O_7_, novel high-temperature MFs, the ferroelectric and magnetic properties of which are still being researched. These structures crystallize into trigonal, orthorhombic, and tetragonal syngonies, respectively, which allows us to trace the relationship between magnetoelectric properties and the symmetry of a structure, and to consider impacts given by crystallographic distortions.

The direct contribution of the distortion into electric polarization and weak ferromagnetism was considered, using BiFeO_3_ as an example. We also analyzed the influence of structural transformation of the Bi_x_R_1−x_FeO_3_ family on their magnetoelectric properties. Calculations showed that the linear magnetoelectric effect, suppressed by the spin-modulated structure in pure BiFeO_3_, becomes allowed due to the symmetry of the new phase in Bi_x_R_1−x_FeO_3_, where it is attributed mainly to rare earth (R) ions.

In the case of the rare-earth orthochromites RCrO_3_, it was shown that the ferroelectric and magnetoelectric properties are due to crystallographic distortions. The displacements of oxygen ions from their positions in the initial perovskite phase results in the emergence of electric dipole moments in the vicinity of the Cr ions, which are coupled with the magnetic moments of the Cr ions. The distortive, ferroelectric, and magnetic order parameters were classified according to irreducible representations of the *Pnma* symmetry group of RCrO_3_, which allows the invariant combinations to be composed between these parameters.

For the first time, a similar symmetry consideration was implemented in the analysis of the Ruddlesden–Popper structures. Taking into account the local symmetry of magnetic ions in the RP unit cell, we introduced the relevant magnetic order parameters and classified them according to the irreducible representation of the *I4/mmm* symmetry group, which describes the tetragonal symmetry of the RP structures. Analysis exemplified on the Ca_3_(Ti_1−x_Mn_x_)_2_O_7_ compounds demonstrated the possibility of realizing MEE in the RP phases containing magnetically active cations and allowed the estimation of the magnetoelectric contribution to the thermodynamic potential.

In conclusion, this research allowed us to compare the magnetoelectric effects for different crystal systems of perovskites and thus design a more meaningful organization of the desired experiments.

## Figures and Tables

**Figure 1 materials-15-00574-f001:**
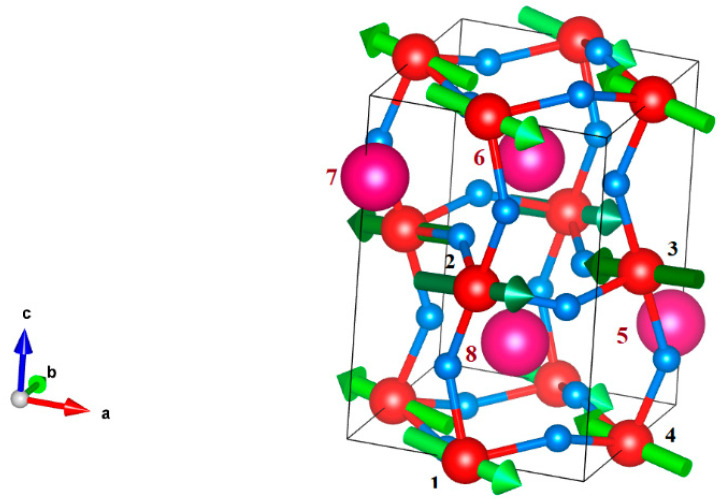
Electric dipole moments arrangement in RCrO_3_ unit cell, a, b, c are the principal crystal axes, Cr ions are numbered as 1, 2, 3, 4; R ions are numbered as 5, 6, 7, 8. Green arrows denote the orientation of electric dipole moments in the vicinity of Cr^3+^ ions ordered by antiferroelectric ***D*** mode.

**Figure 2 materials-15-00574-f002:**
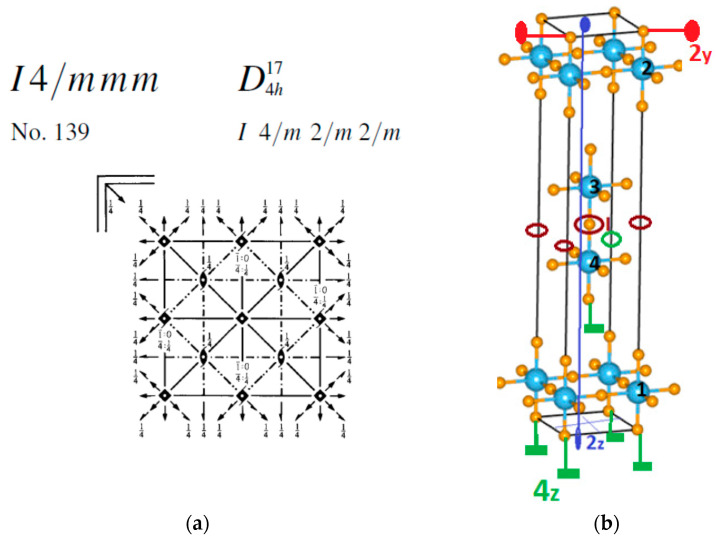
(**a**) *I*4/*mmm* space symmetry group elements; (**b**) Ca_3_Mn_2_O_7_ unit cell, the numbering of magnetic ions is presented, the arrangement of the symmetry elements is also shown-the axes of symmetry 4_z_, 2_z_, 2_y_ and the centers of inversion I.

**Table 1 materials-15-00574-t001:** Irreducible representations of the *Pbnm* symmetry group.

Г_*i*_	$\bar{1}$	2_x_	2_y_	2_z_	Order Parameters, and Magnetic and Electric Fields		
Г_1_	1	1	1	1	*L_y_*		
Г_2_	1	1	−1	−1	*M_x_*	*L_z_*	*H_x_*
Г_3_	1	−1	1	−1	*M_y_*		*H_y_*
Г_4_	1	−1	−1	1	*M_z_*	*L_x_*	*H_z_*
Г_5_	−1	1	1	1	*l_y_*		
Г_6_	−1	1	−1	−1	*l_x_*	*P_z_*	*E_z_*
Г_7_	−1	−1	1	−1		*P_y_*	*E_y_*
Г_8_	−1	−1	−1	1	*l_z_*	*P_x_*	*E_x_*

**Table 2 materials-15-00574-t002:** Irreducible representations of the *Pnma* symmetry group.

Г_*i*_	$\bar{1}$	2_x_	2_y_	2_z_	Magnetic OPs, Magnetic Field		Structural OPs, Electric Field
					4b	4c	
Г_1_	1	1	1	1	$A_{z},G_{x},C_{y}$	*c_y_*	$\Omega_{bx}$
Г_2_	1	1	−1	−1	$F_{z},G_{y},C_{x},H_{z}$	$f_{z},c_{x}$	$\Omega_{by}$
Г_3_	1	−1	1	−1	$F_{x},A_{y},C_{z},H_{x}$	$f_{x},c_{z}$	
Г_4_	1	−1	−1	1	$F_{y},A_{x},G_{z},H_{y}$	*f_y_*	$\Omega_{bz}$
Г_5_	−1	1	1	1		$g_{z},a_{x}$	$Q_{2z}$${,}_{\;}Q_{3x}$, *D_z_*
Г_6_	−1	1	−1	−1		*a_y_*	$P_{z},E_{z}$ $,\;Q_{3y}$
Г_7_	−1	−1	1	−1		$a_{z},g_{x}$	$P_{y},Q_{2x},E_{y}$ _,_ *D_x_*
Г_8_	−1	−1	−1	1		*g_y_*	$P_{x},Q_{2y},E_{x}$ _,_ *D_y_*

**Table 3 materials-15-00574-t003:** Permutation transformations ions in positions 4e {4mm}.

G_F_	1	2	3	4
$\bar{1}$	2	1	4	3
4_z_	1	2	3	4
2_z_	3	4	1	2
2_y_	2	1	4	3

**Table 4 materials-15-00574-t004:** Permutation transformations of basis vectors.

G_F_	F	A	G	C
$\bar{1}$	**F**	−**A**	−**G**	**C**
4_z_	**F**	**A**	**G**	**C**
2_z_	**F**	−**A**	**G**	−**C**
2_y_	**F**	−**A**	−**G**	**C**

**Table 5 materials-15-00574-t005:** Irreducible representations of the *I*4/*mmm* symmetry group and basic functions.

Г_i_	E	$2C_{4z}$	$C_{2z}$	$2C_{2y}$	$\bar{1}$	Basic Vectors *F, G, P*
Г_1_	1	1	1	1	1	
Г_2_	1	1	1	−1	1	*F_z_*
Г_3_	1	−1	1	−1	1	
Г_4_	1	−1	1	1	1	
Г_5_	$\left(\begin{array}{cc} 1 & 0\\ 0 & 1 \end{array}\right)$	$\left(\begin{array}{cc} 0 & 1\\ -1 & 0 \end{array}\right)$	$\left(\begin{array}{cc} -1 & 0\\ 0 & -1 \end{array}\right)$	$\left(\begin{array}{cc} -1 & 0\\ 0 & 1 \end{array}\right)$	$\left(\begin{array}{cc} 1 & 0\\ 0 & 1 \end{array}\right)$	$\left(\begin{array}{c} F_{x}\\ F_{y} \end{array}\right)$
Г_6_	1	1	1	1	−1	*G_z_*
Г_7_	1	1	1	−1	−1	*P_z_*
Г_8_	1	−1	1	−1	−1	
Г_9_	1	−1	1	1	−1	
Г_1__0_	$\left(\begin{array}{cc} 1 & 0\\ 0 & 1 \end{array}\right)$	$\left(\begin{array}{cc} 0 & 1\\ -1 & 0 \end{array}\right)$	$\left(\begin{array}{cc} -1 & 0\\ 0 & -1 \end{array}\right)$	$\left(\begin{array}{cc} -1 & 0\\ 0 & 1 \end{array}\right)$	$\left(\begin{array}{cc} -1 & 0\\ 0 & -1 \end{array}\right)$	$\left(\begin{array}{c} P_{x}\\ P_{y} \end{array}\right)$
${Г}_{10}^{’}$	$\left(\begin{array}{cc} 1 & 0\\ 0 & 1 \end{array}\right)$	$\left(\begin{array}{cc} 0 & 1\\ -1 & 0 \end{array}\right)$	$\left(\begin{array}{cc} -1 & 0\\ 0 & -1 \end{array}\right)$	$\left(\begin{array}{cc} 1 & 0\\ 0 & -1 \end{array}\right)$	$\left(\begin{array}{cc} -1 & 0\\ 0 & -1 \end{array}\right)$	$\left(\begin{array}{c} G_{x}\\ G_{y} \end{array}\right)$

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
