# Peer review of "Symmetry Analysis of Magnetoelectric Effects in Perovskite-Based Multiferroics"

_materials, 2022, doi:10.3390/ma15020574_

Round 1

Reviewer 1 Report

In this article, the authors report a symmetry analysis of the possible magnetoelectric responses in a set of 3 different cases: La doped BiFeO3, Pnma orthochromites RCrO3 and the Ruddlesden-Popper layered perovskites. To do so, the authors used crystallography, symmetries and mixed microscopic/phenomenological order parameters to determine which of the linear magnetoelectric tensor is non-zero and in some cases they give some arguments regarding the amplitudes of these components.
While I agree with the final results (irreps and the alpha tensors), I have several questions/comments regarding some arguments used and I also found several potential errors, either the authors were not careful enough or because the some definitions are indeed not correct or not properly defined.
Hence, the paper cannot be published as it is but it could be reconsidered if the following different points are addressed carefully:

1- (major) The authors define the polarization through Eq. 1. However this expression is a classical point charge definition of the polarization, which is incorrect as the polarization should be estimated through the Berry phase theory and by taking into account the multivalued zero reference (quantum of polarization). However, I understand that an approximation can be done through a sum of the charge times the atomic displacements away from the high symmetry positions but the nominal charge is not the correct charge to be used. Indeed, the Born effective charges should be considered, which are a rank 2 tensor that takes into account that a displacement along x or y can induce a polarization along the z direction (and all the circular permutations). Hence, using a symmetry analysis through a point charge would consider that the Born charge tensor is diagonal, which can give wrong results. I think the authors should justify the use of such a drastic approximation for the polarization and demonstrate that in the crystal structures they analyse they can indeed suppose that considering a Born charges being diagonal is still relevant.

2- (minor) in line 104, the authors mention that the polarization of BiFeO3 can go from 6 to 100 muC/cm^2. I think it should be from 60 to 100 muC/cm^2 and not from 6. Also the large value under epitaxial strain can also be related to the supertetragonal phase induced by large strain.

3- (minor) in the definition of the magnetization, the authors use a single parameter J for the super exchange interaction. This isotropic approximation restricted to the first nearest neighbor should also be justified as the superexchange can be anisotropic (I mean for the different directions, not the Kitaev one, even if it could be at play in some conditions, see e.g. Phys. Rev. B 104, 064431 (2021)) and still present on second nearest neighbors.

4- (medium) In the definitions given in line 159-161, I agree with it but the moment of the La site (l) is actually zero in the case of Lanthanum and so this analysis does not hold if we consider La and a magnetic rare earth, right? This should be stated explicitly that they use the example of La depend BiFeO3 but the analysis is valid for non-zero magnetization on La site as if one places l=0 in Eq.5, then the second matrix on the right gives zero…

5- (medium) In the paragraph from line 200 to line 205, the authors should mention that the high-temperature ferroelectricity reported here is not coming from the rare/earth but from domains (ref. 38, 39 of the manuscript), hence this is not a bulk property. This ferroelectricity in a Pnma phase should not be4 present in pure bulk compound if the rare-earth element is not below its Neel temperature. See, e.g. the review J. Phys.: Condens. Matter 28 (2016) 123001 or Acta Cryst. A74, 308 (2018) and references therein regarding this problem. This can question the analysis done in the present manuscript, which is for the bulk case. 

5- (medium) Still in this RCrO3 analysis section, I would say that the alpha tensor is actually also valid for the RFeO3 crystals as exactly the same configurations are there. I think it should also be mentioned that several of these crystals exhibit non-linear magnetoelectric responses (e.g. GdFeO3,  Nature materials 8, 558 (2009)) and this is not discussed at all in the present manuscript version.

6- (medium) Why the authors do not give the alpha tensor for the Ruddlesden-Popper?

7- (minor) As a general comment I would say that the literature is a bit scarce while several works have been done doing similar analysis, though not through the same view point, e.g. the allowed alpha tensor components are tabulated in the book International Tables of Crystallo Volume D.

Author Response

We thank the Referee for his/her positive evaluation of our manuscript and for valuable comments and advice which helped us in making the presentation stronger.  In response to the referee’s remarks, we have provided point-to-point responses below.

Reviewer-1 comments:

In this article, the authors report a symmetry analysis of the possible magnetoelectric responses in a set of 3 different cases: La doped BiFeO3, Pnma orthochromites RCrO3 and the Ruddlesden-Popper layered perovskites. To do so, the authors used crystallography, symmetries and mixed microscopic/phenomenological order parameters to determine which of the linear magnetoelectric tensor is non-zero and in some cases they give some arguments regarding the amplitudes of these components. While I agree with the final results (irreps and the alpha tensors), I have several questions/comments regarding some arguments used and I also found several potential errors, either the authors were not careful enough or because the some definitions are indeed not correct or not properly defined. Hence, the paper cannot be published as it is but it could be reconsidered if the following different points are addressed carefully:

Remark 1 (major) The authors define the polarization through Eq. 1. However, this expression is a classical point charge definition of the polarization, which is incorrect as the polarization should be estimated through the Berry phase theory and by taking into account the multivalued zero reference (quantum of polarization). However, I understand that an approximation can be done through a sum of the charge times the atomic displacements away from the high symmetry positions but the nominal charge is not the correct charge to be used. Indeed, the Born effective charges should be considered, which are a rank 2 tensor that takes into account that a displacement along x or y can induce a polarization along the z direction (and all the circular permutations). Hence, using a symmetry analysis through a point charge would consider that the Born charge tensor is diagonal, which can give wrong results. I think the authors should justify the use of such a drastic approximation for the polarization and demonstrate that in the crystal structures they analyze they can indeed suppose that considering a Born charges being diagonal is still relevant

Our reply

We thank Referee for this remark. Ferroelectric polarization is currently defined using the Berry phase concept. It is especially convenient for the DFT simulations. In the framework of this concept, polarization has two contributions, one of which comes from nuclei, and the other from electron shells. In fact, the electronic contribution is associated with covalent bonds, which are small in the materials under consideration. In this case, we can restrict ourselves to the first contribution, especially since we are interested in a qualitative analysis and symmetry. The rigorous quantomechanical calculations of the ferroelectric polarization of BiFeO3 were performed in Ref. [Spaldin, N. A. J. Solid State Chem. 2012, 195, 2–10], which gives polarization values consistent with our results. We have taken these remarks into account in the revised manuscript, and we also referred to Ref. [H. Dixit, C. Beekman, C. M. Schlepütz, W. Siemons, Y. Yang, N. Senabulya, R. Clarke, M. Chi, H. M. Christen, and V. R. Cooper, Advanced Science 2, 1500041 (2015).], where diagonal components of effective Born charges of BiFeO3 phases have been calculated. The made changes are as follows:

“The rigorous estimation of electronic polarization requires implementation of the quantum mechanics including electronic structure methods corresponding to experimentally measurable observables [36]. In the frame of this approach in Ref. [37] electric polarization in perovskite BiFeO3 was calculated as a function of percentage distortion from the high symmetry non-polar structure to the ground state R3c structure, which gives the value 95.0 μC cm−2. However, as was shown in Ref. [30] the similar results can be also obtained with the use of the point charge model where the electric polarization is represented through the atomic displacements of Bi and O ions…”

Remark 2  (minor) in line 104, the authors mention that the polarization of BiFeO3 can go from 6 to 100 muC/cm^2. I think it should be from 60 to 100 muC/cm^2 and not from 6. Also, the large value under epitaxial strain can also be related to the supertetragonal phase induced by large strain.

Our reply

Actually, the range of polarization values in BiFeO3 films is in - between 60 - 100 muC/cm^2 [D. Lebeugle, D. Colson, A. Forget, and M. Viret, Appl. Phys. Lett. 91, 022907 (2007)], however, in ceramic samples, the polarization can attain much lower values, which can be of the order 8.9 muC/cm^2 [Y. P. Wang, G. L. Yuan, X. Y. Chen, J.-M. Liu, and Z. G. Liu, J. Phys. D: Appl. Phys. 39, 2019 (2006) ] at room temperature as well as in single crystals, where P ~6. muC/cm^2 at 77 K [J. R. Teague, R. Gerson, and W. J. James, Solid State Communications 8, 1073 (1970).]

We added these references in the revised manuscript and rewrote the paragraph as follows:

“The magnitude of polarization has been the subject of controversy for a while. As was reported in Ref. [5,20,31–35] spontaneous polarization in BiFeO3 crystal and films can attain the values varying from 6 to 150 μC/cm2. Electric polarization is sufficiently low Ps ~ 6- 9 μC/cm2 in single crystals and ceramic samples [34,35], the large values of Ps, being around 100 μC/cm2, are achieved under epitaxial strain in the films [5,36–38], they can also be related to the supertetragonal phase induced by the strain.”

Remark 3  - (minor) in the definition of the magnetization, the authors use a single parameter J for the superexchange interaction. This isotropic approximation restricted to the first nearest neighbor should also be justified as the superexchange can be anisotropic (I mean for the different directions, not the Kitaev one, even if it could be at play in some conditions, see e.g. Phys. Rev. B 104, 064431 (2021)) and still present on second nearest neighbors.

Our reply

            We thank Referee for the interesting question. Accounting for the nearest neighbor contribution is an interesting and important problem, but since we are focusing on symmetry, the assumption cannot be considered as a limiting factor.  

The assumption that the Heisenberg exchange constant remains unchanged is commonly used approximation for multiferroic BiFeO3. Deviation (non – collinearity) of the nearest neighbor spins, in particular, can be accounted by the antisymmetric Dzyaloshinskii – Moriya exchange interaction [R. S. Fishman, T. Rõõm, and R. de Sousa, Phys. Rev. B 99, 064414 (2019), Zvezdin, A. K.; Pyatakov, A. P. EPL Europhys. Lett. 2012, 99 (5), 57003.], we added these references in the revised manuscript. Authors of  Ref. [ Phys. Rev. B 104, 064431 (2021)] report on the orthorhombic multiferroics (orthoferrites) whose symmetry differs from BiFeO3 symmetry. We refer to this paper later in Section 2.2 where we consider orthochromites. The made changes are following

“Here we used an assumption that the Heisenberg exchange constant remains unchanged, which is commonly used approximation for multiferroic BiFeO3, in particular, deviation (non – collinearity) of the nearest neighbor spins is accounted by the antisymmetric Dzyaloshinskii – Moriya exchange interaction as was described in Refs. [42,43].”

Remark 4  - (medium) In the definitions given in line 159-161, I agree with it but the moment of the La site (l) is actually zero in the case of Lanthanum and so this analysis does not hold if we consider La and a magnetic rare earth, right? This should be stated explicitly that they use the example of La depend BiFeO3 but the analysis is valid for non-zero magnetization on La site as if one places l=0 in Eq.5, then the second matrix on the right gives zero…

Our reply

We thank Referee for this comment, we made the corrections in the revised manuscript, emphasizing that the performed analysis is valid to describe the magnetoelectric effect in Bi1-xRxFeO3 compounds for the case of non –zero magnetization on the R ion site, for example for R=Gd, Dy, …, while in the case R=La in order to understand magnetoelectricity it is necessary to appeal to structural order parameters as in Section 2.2.

Remark 5- (medium) In the paragraph from line 200 to line 205, the authors should mention that the high-temperature ferroelectricity reported here is not coming from the rare/earth but from domains (ref. 38, 39 of the manuscript), hence this is not a bulk property. This ferroelectricity in a Pnma phase should not be present in pure bulk compound if the rare-earth element is not below its Neel temperature. See, e.g. the review J. Phys.: Condens. Matter 28 (2016) 123001 or Acta Cryst. A74, 308 (2018) and references therein regarding this problem. This can question the analysis done in the present manuscript, which is for the bulk case.

Our reply

We thank Referee for the valuable review papers important to be referred  in the manuscript, we added these papers to the reference list and discuss them in the Introduction.

However, we did not write that the approach presented in Sec.2.2 is used to describe high–temperature ferroelectricity. Ferroelectric properties of rare earth orthochromites manifest themselves below Neel temperature, see e.g. GdCrO3 [J. Phys.: Condens. Matter 28 (2016) 123001], in this case, the classification presented in Table 2 is commonly applied to describe the magnetoelectric effect in these compounds, please see also Refs.  [A. K. Zvezdin and A. A. Mukhin, Jetp Lett. 88, 505 (2008),  E. A. Turov, Phys.-Usp. 37, 303 (1994).]

Remark 6  - (medium) Still in this RCrO3 analysis section, I would say that the alpha tensor is actually also valid for the RFeO3 crystals as exactly the same configurations are there. I think it should also be mentioned that several of these crystals exhibit non-linear magnetoelectric responses (e.g. GdFeO3,  Nature materials 8, 558 (2009)) and this is not discussed at all in the present manuscript version.

Our reply

We thank Referee for this comment, at first we aimed to consider RCrO3 in Sec.2.2, but for sure this consideration can be expanded to RFeO3 crystals. We wrote  

"At the end of this section, we draw the reader's attention to the fact that the presented analysis applies to rare-earth orthoferrites RFeO3. RFeO3 compounds crystallize in the same space symmetry group Pnma, and several of these crystals exhibit non-linear magnetoelectric responses [10,10–13]. The magnetoelectric tensor (12) can be also used to describe their magnetoelectric properties. " 

Remark 7  - (medium) Why the authors do not give the alpha tensor for the Ruddlesden-Popper?

Our reply

We calculated the magnetoelectric tensor for Ruddlesden – Popper structure, which is given by formula (16) in the revised manuscript.

Remark 8  - (minor) As a general comment I would say that the literature is a bit scarce while several works have been done doing similar analysis, though not through the same view point, e.g. the allowed alpha tensor components are tabulated in the book International Tables of Crystallo Volume D.

Our reply

We thank Referee for helpful and valuable recommendations. In the revised manuscript, we rewrote parts of the introduction and added references, including those recommended by the Referee, to the bibliography. In total, we added 21 references among which are the following:

(23)     Senn, M. S.; Bristowe, N. C. A Group-Theoretical Approach to Enumerating Magnetoelectric and Multiferroic Couplings in Perovskites. Acta Crystallogr. Sect. Found. Adv. 2018, 74 (4), 308–321. https://doi.org/10.1107/S2053273318007441.

(24)     Mulder, A. T.; Benedek, N. A.; Rondinelli, J. M.; Fennie, C. J. Turning ABO3 Antiferroelectrics into Ferroelectrics: Design Rules for Practical Rotation-Driven Ferroelectricity in Double Perovskites and A3B2O7 Ruddlesden-Popper Compounds. Adv. Funct. Mater. 2013, 23 (38), 4810–4820. https://doi.org/10.1002/adfm.201300210.

(25) Bousquet, E.; Cano, A. Non-Collinear Magnetism in Multiferroic Perovskites. J. Phys. Condens. Matter 2016, 28 (12), 123001. https://doi.org/10.1088/0953-8984/28/12/123001.

(26)     Narayanan, N.; Graham, P. J.; Rovillain, P.; O’Brien, J.; Bertinshaw, J.; Yick, S.; Hester, J.; Maljuk, A.; Souptel, D.; Büchner, B.; Argyriou, D.; Ulrich, C. Reduced Crystal Symmetry as Origin of the Ferroelectric Polarization within the Incommensurate Magnetic Phase of TbMn2O5. ArXiv210905164 Cond-Mat 2021.

(27)     Perez-Mato, J. M.; Ribeiro, J. L.; Petricek, V.; Aroyo, M. I. Magnetic Superspace Groups and Symmetry Constraints in Incommensurate Magnetic Phases. J. Phys. Condens. Matter 2012, 24 (16), 163201. https://doi.org/10.1088/0953-8984/24/16/163201.

(28)     Hatch, D. M.; Stokes, H. T. INVARIANTS: Program for Obtaining a List of Invariant Polynomials of the Order-Parameter Components Associated with Irreducible Representations of a Space Group. J. Appl. Crystallogr. 2003, 36 (3), 951–952. https://doi.org/10.1107/S0021889803005946.

(29)     (International Tables for Crystallography) Introduction to the properties of tensors https://onlinelibrary.wiley.com/iucr/itc/Da/ch1o1v0001/ (accessed 2021 -12 -27).

(31)     Palkar, V. R.; John, J.; Pinto, R. Observation of Saturated Polarization and Dielectric Anomaly in Magnetoelectric BiFeO3 Thin Films. Appl. Phys. Lett. 2002, 80 (9), 1628–1630. https://doi.org/10.1063/1.1458695.

(33)     Lebeugle, D.; Colson, D.; Forget, A.; Viret, M. Very Large Spontaneous Electric Polarization in BiFeO3 Single Crystals at Room Temperature and Its Evolution under Cycling Fields. Appl. Phys. Lett. 2007, 91 (2), 022907. https://doi.org/10.1063/1.2753390.

(34)     Teague, J. R.; Gerson, R.; James, W. J. Dielectric Hysteresis in Single Crystal BiFeO3. Solid State Commun. 1970, 8 (13), 1073–1074. https://doi.org/10.1016/0038-1098(70)90262-0.

(35)     Wang, Y. P.; Yuan, G. L.; Chen, X. Y.; Liu, J.-M.; Liu, Z. G. Electrical and Magnetic Properties of Single-Phased and Highly Resistive Ferroelectromagnet BiFeO3ceramic. J. Phys. Appl. Phys. 2006, 39 (10), 2019–2023. https://doi.org/10.1088/0022-3727/39/10/006.

(36)     Li, J.; Wang, J.; Wuttig, M.; Ramesh, R.; Wang, N.; Ruette, B.; Pyatakov, A. P.; Zvezdin, A. K.; Viehland, D. Dramatically Enhanced Polarization in (001), (101), and (111) BiFeO3 Thin Films Due to Epitiaxial-Induced Transitions. Appl. Phys. Lett. 2004, 84 (25), 5261–5263. https://doi.org/10.1063/1.1764944.

(37)     Dixit, H.; Beekman, C.; Schlepütz, C. M.; Siemons, W.; Yang, Y.; Senabulya, N.; Clarke, R.; Chi, M.; Christen, H. M.; Cooper, V. R. Understanding Strain-Induced Phase Transformations in BiFeO3 Thin Films. Adv. Sci. 2015, 2 (8), 1500041. https://doi.org/10.1002/advs.201500041.

(38)     Sando, D.; Agbelele, A.; Rahmedov, D.; Liu, J.; Rovillain, P.; Toulouse, C.; Infante, I. C.; Pyatakov, A. P.; Fusil, S.; Jacquet, E.; Carrétéro, C.; Deranlot, C.; Lisenkov, S.; Wang, D.; Le Breton, J.-M.; Cazayous, M.; Sacuto, A.; Juraszek, J.; Zvezdin, A. K.; Bellaiche, L.; Dkhil, B.; Barthélémy, A.; Bibes, M. Crafting the Magnonic and Spintronic Response of BiFeO3 Films by Epitaxial Strain. Nat. Mater. 2013, 12 (7), 641–646. https://doi.org/10.1038/nmat3629.

(39)     Resta, R.; Vanderbilt, D. Theory of Polarization: A Modern Approach. In Physics of Ferroelectrics: A Modern Perspective; Topics in Applied Physics; Springer: Berlin, Heidelberg, 2007; pp 31–68. https://doi.org/10.1007/978-3-540-34591-6_2.

(40)     Spaldin, N. A. A Beginner’s Guide to the Modern Theory of Polarization. J. Solid State Chem. 2012, 195, 2–10. https://doi.org/10.1016/j.jssc.2012.05.010.

and other references.

Reviewer 2 Report

The article discusses how the symmetry of a crystal structure affects the magnetoelectric effect in meltiferroic materials such as Bi1-xLaxFeO3, RCrO3, Ca3(Ti1-xMnx)2O7, using crystallographic distortion as the primary order parameters. The approach is sound and interesting. However, the article is somewhat difficult to follow. Please see my comments/questions below.

  1. In the introduction, giving an overview of the research subject is always nice for someone who is new to the field. It would have been more helpful if the authors provided the background information on what we do or don't currently understand about the multiferroic effect and why the approach taken in this work was unique.
  2. The paper (arXiv:2109.05164) that appeared on arXiv makes a similar argument with RMn2O5. The authors may consider citing this paper to let the readers know that the symmetry argument is also being actively pursued by others as well.
  3. On page 4, there are three expressions right above Table 1. Gamma needs to be defined. Please provide at least a brief description of each expression.
  4. The tables in the article are difficult to follow. Please make them more readable. 
  5. On page 5, where does the first sentence start?
  6. On page 5 line 203, do the authors mean "precess", not "process"?  

Author Response

Reviewer 2

We thank the Referee for his/her positive evaluation of our manuscript and for the valuable comments which helped us to improve the manuscript.

Reviewer-2 comments:

The article discusses how the symmetry of a crystal structure affects the magnetoelectric effect in multiferroic materials such as Bi1-xLaxFeO3, RCrO3, Ca3(Ti1-xMnx)2O7, using crystallographic distortion as the primary order parameters. The approach is sound and interesting. However, the article is somewhat difficult to follow. Please see my comments/questions below

Remark 1

In the introduction, giving an overview of the research subject is always nice for someone who is new to the field. It would have been more helpful if the authors provided the background information on what we do or don't currently understand about the multiferroic effect and why the approach taken in this work was unique.

Our reply

We thank Referee for this comment. We add a paragraph in the Introduction explaining the purpose of our research and the difference (uniqueness) of our approach from the works of other authors.

We wrote “It is valuable to note that the classification of distortions and the search for recipes for effective magnetoelectric couplings in MFs with a perovskite structure is a long-standing and, at the same time, “hot” problem. We refer to several reviews, references therein, and original papers that discussed various classification schemes for perovskite distortions (Glazer, Aleksandrov & Bartolome, and Fennie) and group – theoretical approaches, including their implementation in online tools (in particular, ones on the Bilbao Crystallographic Server) for identification of active order parameters, their possible couplings and invariant polynomials [22–29]. However, distortion classification schemes mainly deal with BO6 octahedrons and do not account for polar distortions, which give a significant impact on ferroelectric and MF properties, in addition, they also have limitations for specific systems. Despite the power of the software tools, they have not yet been applied to each MF system, for example, the details of the magnetoelectric coupling in MFs with a variable concentration of rare-earth ions have yet not been investigated, same statement is also applied to the R-P structures.

So, in our research, we appeal to group-theoretical analysis, which is an elegant and effective tool for studying the properties of crystals with a complex magnetic structure, to classify the distortive, ferroelectric, and magnetic orderings in several classes of perovskite MFs. So, in our research, we appeal to group-theoretical analysis, which is an elegant and effective tool for studying the properties of crystals with a complex magnetic structure, to classify the distortive, ferroelectric, and magnetic orderings in several classes of perovskite MFs.”

Remark 2  

The paper (arXiv:2109.05164) that appeared on arXiv makes a similar argument with RMn2O5. The authors may consider citing this paper to let the readers know that the symmetry argument is also being actively pursued by others as well

Our reply

We thank Referee for the comment and reference; we added this reference to the reference list as Ref. [29] and discuss it in the Introduction

Remark 3  

On page 4, there are three expressions right above Table 1. Gamma needs to be defined. Please provide at least a brief description of each expression.

Our reply

According to the Referee suggestion, in the revised version of the manuscript, we wrote the formula for magnetoelectric energy (5), which combined these three expressions and added the definition of gamma (magnetoelectric coefficient).

Remark 4  

The tables in the article are difficult to follow. Please make them more readable.

Our reply

The Tables are presented in the format required for the journal, we added blank spaces between the variables to make them more readable.

Remark 5  

On page 5, where does the first sentence start?

Our reply

We thank Referee for noticing this misprint, we rewrote the first sentence on page 5 as follows

“Magnetoelectric coupling coefficient for the Bi1-xRxFeO3 structures can be determined as follows ”

Remark 6

On page 5 line 203, do the authors mean "precess", not "process"?    

Our reply

No, the meaning is namely the processes of spin – reorientation,  we rewrote this collocation as “spin reorientation phase transitions” to avoid misunderstanding.

Round 2

Reviewer 1 Report

The authors went through all of my comments and gave a very clear and extended response to them. The list of publications has been strongly enlarged in the new manuscript and all the key detailed point I was asking for in my comments are now addressed in the new manuscript.

I think that the paper can be accepted and I thank the authors for their response quality.